# Generalized News Event Discovery via Dynamic Augmentation and Entropy Optimization

## ABSTRACT

News event discovery refers to the identification and detection of news events using multimodal data on social media. Currently, most works assume that the test set consists of known events. However, in real life, the emergence of new events is more frequent, which invalidates this assumption. In this paper, we propose a Dynamic Augmentation and Entropy Optimization (DAEO) model to address the scenario of generalized news event discovery, which requires the model to not only identify known events but also distinguish various new events. Specifically, we first introduce a multimodal augmentation module, which utilizes adversarial learning to enhance the multimodal representation capability. Secondly, we design an adaptive entropy optimization strategy combined with a self-distillation method, which uses multi-view pseudo-label consistency to improve the model's performance on both known and new events. In addition, we collect a multimodal news event discovery (MNED) dataset of 161,350 samples annotated with 66 real-world events. Extensive experimental results on the MNED dataset demonstrate the effectiveness of our proposed method. Our dataset is available on https://anonymous.4open.science/r/2FF5.

## CCS CONCEPTS

• **Information systems** → **Multimedia information systems**; • **Computing methodologies** → **Artificial intelligence**.

## KEYWORDS

Generalized News Event Discovery, Social Media

## 1 INTRODUCTION

News event discovery aims at automatically identifying and classifying news events from a wide range of data sources. In the age of social media, it becomes particularly crucial, as platforms like Twitter, Facebook, and others have emerged as primary channels for news dissemination. Compared to traditional media, news spreads faster and reaches a wider audience on social media, making the timely discovery and tracking of news events even more vital. Applications of news event discovery span various domains, including crisis management [23], public sentiment monitoring [18], market analysis [27], and public safety [17]. In these contexts, accurate and prompt identification of news events can help organizations and

**Unpublished working draft. Not for distribution.**
Permission to make digital or hard copies of all or part of this work for personal or classroom use is granted without fee provided that copies are not made or distributed for profit or commercial advantage and that copies bear this notice and the full citation on the first page. Copyrights for components of this work owned by others than the author(s) must be honored. Abstracting with credit is permitted. To copy otherwise, or republish, to post on servers or to redistribute to lists, requires prior specific permission and/or a fee. Request permissions from permissions@acm.org.
*ACM MM, 2024, Melbourne, Australia*
© 2024 Copyright held by the owner/author(s). Publication rights licensed to ACM.
ACM ISBN 978-x-xxxx-xxxx-x/YY/MM
https://doi.org/10.1145/nnnnnnn.nnnnnnn

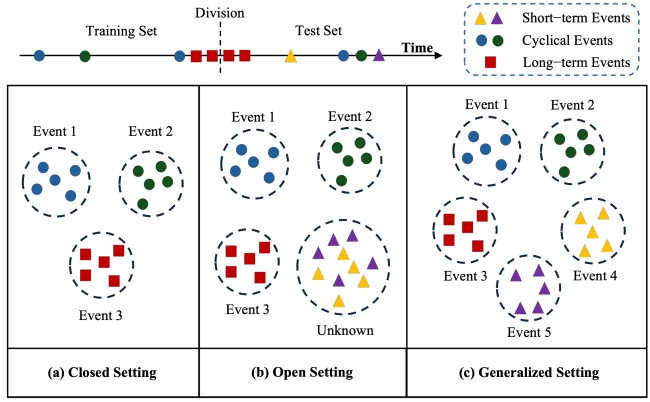

**Figure 1: Different settings for news event discovery. Events 4 and 5 are new events that do not occur in the training set.**

individuals respond more swiftly, better understand and analyze public opinion, and manage information flow more effectively.

The current research [1, 13, 14, 36, 40] in news event discovery primarily focuses on utilizing multimodal approaches due to the richer information provided by the multimodal data. However, a significant limitation of these studies is their reliance on a closed-set assumption, which greatly diminishes their applicability in practical scenarios. As illustrated in Figure 1a, under a closed setting, the focus is mainly on identifying news events that are already known, like cyclical or long-term events, which have happened before. This kind of approach falls short when it comes to detecting novel events that emerge over time. In response to this limitation, some researchers [24] have proposed shifting towards an open setting, aiming to identify novel events as they occur as shown in Figure 1b. Yet, as the volume of new events grows in real life, merely distinguishing whether an event is unknown or not is often insufficient. This has led to the exploration of the generalized news event discovery problem, which seeks to extend beyond the binary classification of new events as either known or unknown. This task aims at not only recognizing previously occurred news events but also differentiating among events that have not yet occurred, referring to this capability as general category discovery [30]. As shown in Figure 1c, it requires the model to both identify known events and categorize new events.

However, the task of generalized news event discovery presents several challenges. The first challenge lies in dealing with multimodal features. When identifying news events, it's possible that only one modality, either text or image, provides useful information, or both modalities offer complementary insights. This variability requires the effective integration and utilization of each modality, especially for events that are closely related or similar in nature. The second challenge involves utilizing knowledge of previously

occurred events to identify new events. This necessitates a model capable of distinguishing subtle differences between known and new events. Lastly, the challenge of dataset scarcity compounds the difficulty of this task. A comprehensive dataset, rich in both volume and variety of events, including temporal information, is crucial for this task. As illustrated in Figure 1, temporal information plays a crucial role in the division of datasets. Unfortunately, existing datasets [14, 24] lack this temporal metadata, leading to the use of random splits for training and test sets, which can not reflect the real-world scenario where events unfold over time.

To address the aforementioned challenges, we introduce a Dynamic Augmentation and Entropy Optimization (DAEO) model. For the first challenge, we design a multimodal augmentation module to learn more robust multimodal event features, which implicitly leverages the label information of events to learn the relationship between different modalities. It utilizes adversarial learning to not only encourage the generation of multimodal features that can distinguish between different similar events but also ensure the generated features are as diverse as possible. For the second challenge, we learn a unified prototypical classification head for all new and known classes with self-distillation learning. Unlike previous methods [32] that used entropy maximization for all samples, we introduce an adaptive entropy optimization technique. Specifically, we generate various pseudo labels using a multi-view approach, including single-modal random augmentations (e.g., image augmentations) and outputs from the multimodal augmentation module. Then, when there is consistency across multiple views, the model is optimized to minimize entropy, thereby enhancing confidence in identified known events. Conversely, when views differ, entropy maximization is employed to encourage further exploration of the new events. Furthermore, we collect a multimodal news event discovery (MNED) dataset for generalized news event discovery from Twitter, comprising 161,350 multimodal samples annotated with 66 real-world events. Reflecting the temporal characteristics of news events, we define and collect three types of news events: short-term, cyclical, and long-term events. To ensure diversity, each event type encompasses a broad range of sub-events, including short-term events like natural and man-made disasters, terrorist attacks; cyclical events such as sporting events, political elections, international summits; and long-term events covering political conflicts, economic/social crises, and environmental/health issues.

The contributions of this paper can be summarized as follows:

- We formulate the task of generalized news event discovery and introduce a Dynamic Augmentation and Entropy Optimization (DAEO) model designed to tackle this task.
- We propose a multimodal augmentation module and an adaptive entropy optimization strategy aimed at improving the representation of multimodal features and enhancing the ability to uncover new events, respectively.
- We collect a comprehensive multimodal news event discovery (MNED) dataset deigned for news event discovery, which encompasses a wide array of events categorized into long-term, cyclical, and short-term events, providing a rich resource for the research community.
- Extensive experimental results on the MNED dataset demonstrate the effectiveness of our proposed model.

## 2 RELATED WORK

### 2.1 News Event Discovery

The exploration of news event discovery initially stemmed from the domain of topic detection and tracking [2]. With the rise of the internet, an increasing number of researchers have shifted their focus towards utilizing single-modal information present on social media, such as images or text, to facilitate the discovery of news events [12, 21, 22, 26, 28, 38]. For instance, Zaharieva et al. [38] utilized the image information for the detection of specific social events. Lee et al. [12] employed a naive Bayes multinomial classifier for identifying distinct trending topics. Despite their contributions, these single-modal approaches face inherent limitations when compared to analyses that incorporate multimodal data, as accurately detecting news events often requires a synthesis of various event-related elements. Therefore, researchers began to delve into news event discovery based on multimodal data [13, 14, 19, 33–35, 37]. For example, Li et al. [13] proposed a transformer-based conditioned variational autoencoder to jointly model the textual information, visual information and label information for incomplete social event classification. Lin et al. [14] designed a multi-modal fusion with external knowledge to address the out-of-distribution issue in news event detection. These approaches, however, assume that the training and test events remain consistent (i.e., a closed set), which diminishes their efficacy in the face of new, unseen events. Recently, Qian et al. [24] introduced an open-world social event classifer model towards an open setting of news event discovery, which can not only distinguish already occurred events but also identify whether an event is unknown when new events emerge. Nonetheless, the practicality of this approach faces challenges as the number of new events continuously increases over time, making the simple binary distinction between whether an event is new or not insufficient.

### 2.2 Generalized Category Discovery

The field of generalized category discovery has recently emerged, focusing on classifying known categories while also identifying unseen, new categories. The pioneering work in this domain was conducted by Vaze et al. [30], who introduced the idea of leveraging a universal feature representation to discover new categories. Specifically, they proposed fine-tuning a pre-trained DINO ViT [6] using a combination of one supervised and one self-supervised contrastive method. This approach is further complemented by a semi-supervised clustering for label assignment. In addition, the authors extended UNO [7] and RankStats [8] for this task, which were originally designed for novel class discovery [7, 39]. However, these methods employ a two-step training process, involving feature learning and clustering, which could potentially be sub-optimal. To address this, Wen et al. [32] suggested parametric approaches that construct a trainable classifier, enabling the joint optimization of the entire network. Similar to the idea behind DINO ViT [6], their method used the generation of pseudo cluster labels to guide the learning of new categories. This work sparked a series of follow-up studies [20, 31]. For example, Wang et al. [31] proposed the use of CLIP-generated text to guide image learning for category discovery. Nevertheless, it is challenging to apply these methods directly to generalized news event discovery, which involves multimodal

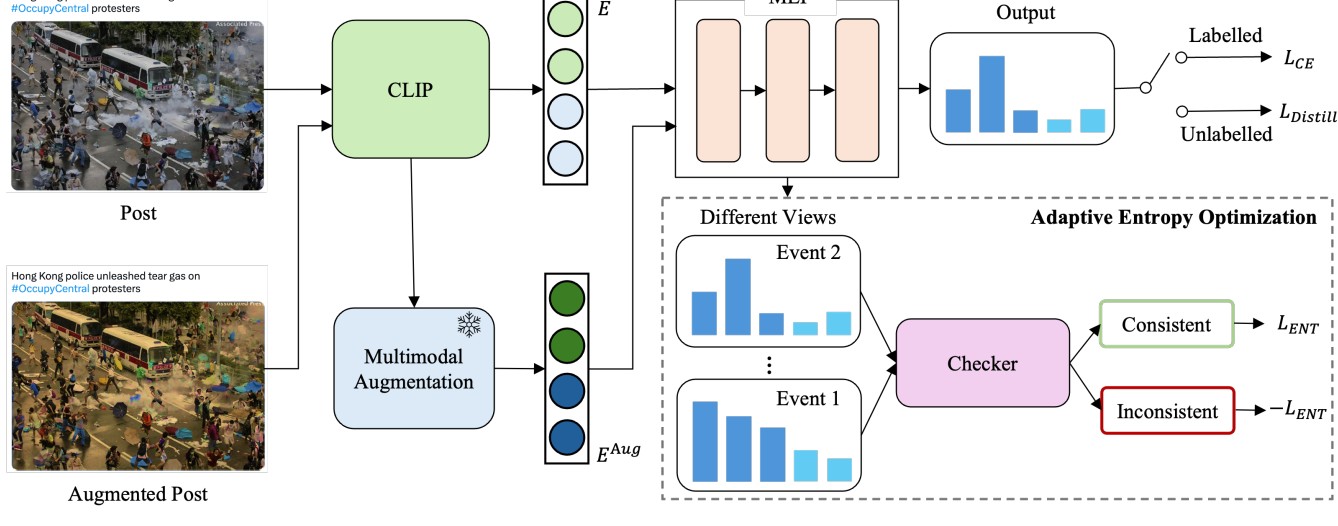

**Figure 2: The framework of the proposed Dynamic Augmentation and Entropy Optimization (DAEO) model.**

data and higher-level event labels. This is because these methods rely on image data and pre-trained models developed primarily for similar tasks, which emphasizes the need for specialized adaptation in generalized news event discovery.

## 3 PRELIMINARIES AND PROBLEM STATEMENT

**Problem 1 (Generalized News Event Discovery).** Given a dataset $D$ contains two parts: $D_L$ containing known events and $D_U$ including both known and new events, organized chronologically. A model is expected to be developed that can accurately categorize both known and new events in $D_U$.

More specifically, $D_L = \{(x_i, y_i)\}_{i=1}^N$ constitutes a labeled dataset containing multimodal instances $x_i$, each labeled with $y_i$ from the set $Y_L$ of known event categories. $D_U = \{(x_j)\}_{j=1}^M$ represents an unlabeled dataset with multimodal instances $x_j$, which are to be associated with labels from an expanded set $Y_U$. The set $Y_U$ includes new, unseen event categories denoted by $Y_{new}$, and a subset of $Y_L$, designated as $Y_{future}$, which represents a subset of $Y_L$ that will continue to happen in the future. Hence, the relationship $Y_U = Y_{new} \cup Y_{future}$, with $Y_{future} \subseteq Y_L$ as not all events from $Y_L$ are expected to reoccur. During training, the model is concurrently trained on both $D_L$, to learn from the historical occurrence of events, and $D_U$, to anticipate and categorize future, unseen events.

In addition, in order to ensure that there are enough event types and relationships that can be used for generalized news event discovery, we define three types of news events based on their temporal attributes: short-term, cyclical, and long-term events. The following are the formal definitions:

**Definition 1: (Short-term Event).** A short-term event is characterized by its ephemeral nature, typically unfolding and concluding within a brief time span. Examples of such events include natural disasters, sudden political upheavals, or unexpected public incidents. These events are transient and unpredictable, hence they

have a high probability of falling into both $Y_L \setminus Y_{future}$ (elements present in $Y_L$ but absent in $Y_{future}$) and $Y_{new}$ since they may not have occurred in the past or might represent entirely new scenarios.

**Definition 2: (Cyclical Event).** Cyclical events are those that occur at regular intervals, marked by their predictability and periodicity. An example of a cyclical event is the Olympic Games, which recur on a four-year cycle. These events are anticipated and are typically encompassed within $Y_{future}$ due to their recurrent nature.

**Definition 3: (Long-term Event).** Long-term events span extended periods, often unfolding over months, years, or even decades. Wars, economic recessions, or major policy reforms are examples of long-term events. These events persist over such durations that they may be present in both $Y_L$ and $Y_U$.

## 4 METHODOLOGY

### 4.1 Overview of the Framework

As illustrated in Figure 2, our DAEO model begins by leveraging a pretrained CLIP model [25] to extract features from both images and texts, which are then concatenated to form multimodal event features $E$. Specifically, to enable self-learning from unlabeled data, we apply random data augmentation to the images of the input posts to obtain an augmented post for distillation learning. The multimodal augmentation module then employs adversarial learning to generate robust multimodal augmented features $E^{Aug}$, which enhances the classifier's ability to distinguish between similar events. Then, we adopt a multilayer perceptron (MLP) as classifier $f$ to obtain the output. For labeled data, we employ standard supervised learning techniques using the labels; for unlabeled data, we utilize distillation learning for training. Additionally, the adaptive entropy optimization module uses the generated multi-view pseudo-labels for consistency checking to selectively optimize entropy. This approach not only encourages the discovery of new events but also improves the accuracy of known events.

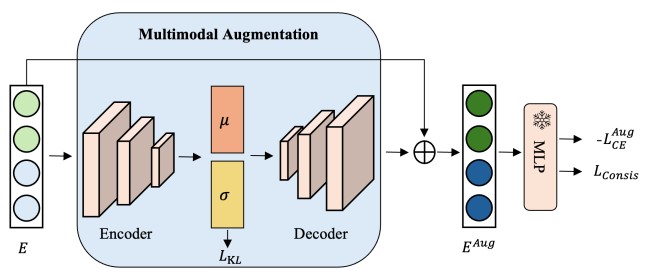

**Figure 3: Multimodal augmentation module.**

## 4.2 Multimodal Event Feature Extraction

According to [30], it is crucial to adopt a robust pretrained model to discover new category, like DINO ViT [6]. However, most pretrained models are primarily focused on image data. Thanks to the cross-modal alignment training on very large-scale image-text pairs, CLIP [25] demonstrates strong zero-shot performance, evidencing its powerful generalization capability for multi-modal joint embedding. Therefore, given an input sample $x_i$, we utilize the pretrained CLIP ViT-B/16 model to generate features for both the images and texts. These features are then concatenated to form our multimodal event feature $E_i$, which can be represented as:

$$E_i = CLIP(x_i). \tag{1}$$

## 4.3 Multimodal Augmentation

In generalized news event discovery, it is important for a model to distinguish similar news events finely, such as different earthquakes in disaster events. Previous methods [30] utilize the supervised contrastive learning and self-contrastive learning method to widen the decision margins between different categories. However, applying random data augmentation for contrastive learning on single modalities, such as text or images, does not seem to enhance model performance for multimodal data (see Sec. 5.6). A possible reason is that random augmentation, especially for text, might lead to the loss of event-related clues, causing negative optimization. For news event discovery tasks, the relationship between images and text can be complementary, related, or unrelated.

In our model, we adopt a different approach to learn robust features, i.e., the multimodal augmentation module, by generating multimodal augmented features through the adversarial method [15] at the feature level. On one hand, we aim for the generated multimodal augmented features to closely approach the decision boundary, which improves the classifier's ability to distinguish between similar events. On the other hand, we strive to ensure that the generated multimodal features retain the original event semantics, which prevents negative optimization.

As shown in Figure 3, we employ a Variational Autoencoder (VAE) model [10] as the generative model, denoted as $G$, which includes an encoder and a decoder. The VAE model has been proven effective in generating features. We utilize the KL divergence [9] to make the encoder's output as close to a standard Gaussian distribution as possible, which can be formulated as:

$$L_{KL} = -\frac{1}{2N}\sum_{i=1}^{N}\left(1+\log(\sigma_i^2)-\mu_i^2-\sigma_i^2\right) \tag{2}$$

where $\mu$ and $\sigma$ are the mean and standard deviation parameters output by the encoder, respectively. Furthermore, we use a residual module to retain more of the original multimodal feature semantics. The augmented features $E_i^{Aug}$ can be formulated as:

$$E_i^{Aug} = G(E_i) + E_i. \tag{3}$$

For learning to generate robust multimodal augmented features, we perform the adversarial training consisting of two parts. In the first part, as shown in Figure 2, we fix the parameters of the multimodal augmentation module $G$ and train the CLIP and classifier model $f$ to minimize the cross-entropy loss between the output and the true event labels, which ensures that augmented features retain their original semantics. It can be formulated as:

$$L_{CE}^{Aug} = -\frac{1}{N}\sum_{i=1}^{N}\ell_{ce}(f(E_i^{Aug}), y_L^{(i)}), \tag{4}$$

where $\ell_{ce}(\cdot)$ represents the cross-entropy loss function.

In the second part, as shown in Figure 3, we fix the parameters of the CLIP and classifier models and train $G$, on one hand, maximize the cross-entropy loss between the output and the true event labels to generate more discriminative features closed to the decision boundary, and on the other hand, minimize the consistency loss to align the semantics of the augmented and original multimodal feature outputs. The consistency loss can be formulated as:

$$L_{Consis} = -\frac{1}{N}\sum_{i=1}^{N}f(E_i)\log(f(E_i^{Aug})). \tag{5}$$

To achieve the adversarial goal, we want the optimal parameters $\hat{\theta}_{CLIP}$, $\hat{\theta}_G$ and $\hat{\theta}_f$ to jointly satisfy

$$(\hat{\theta}_{CLIP}, \hat{\theta}_f) = \arg\min_{\theta_{CLIP},\theta_f} L_{CE}^{Aug} + L_{CE}, \tag{6}$$

$$(\hat{\theta}_G) = \arg\max_{\theta_G} L_{CE}^{Aug} - L_{Consis} - L_{KL}, \tag{7}$$

$$L_{CE} = -\frac{1}{N}\sum_{i=1}^{N}\ell_{ce}(f(E_i), y_L^{(i)}). \tag{8}$$

In this way, the generated features $E_i^{Aug}$ will be close to the decision boundary, which further helps the classifier $f$ to distinguish the class with some ambiguous decision boundaries.

## 4.4 Adaptive Entropy Optimization

To identify new categories, we train a unified prototypical classification head for all new and known classes using a self-distillation framework. For self-distillation, we perform simple augmentations on images to obtain augmented images. Considering the potential for existing random text augmentation methods to change semantics and cause negative optimization, we compose augmented multimodal data directly from augmented images and original texts. Through the CLIP model, we obtain two different views of multimodal features, $E_i$ and $E_i'$. Then, we map these multimodal features to $K$-dimensional vectors as outputs using a function $f$, where $K = |Y_L \cup Y_U|$ is the total number of event categories. For labeled data, we optimize using a cross-entropy function in Eq. 8. For unlabeled data, we employ self-distillation learning. Specifically, we first randomly initialize a set of prototypes $C = \{c_1, ..., c_K\}$, each

representing one category. During training, we compute the cosine similarity between the output features and prototypes to obtain soft labels $p_i/q_i$ for each view, which can be formulated as:

$$p_i^{(k)} = \frac{\exp\left(\frac{1}{\tau}(f(E_i)/\|f(E_i)\|_2)^T(\mathbf{c}_k/\|\mathbf{c}_k\|_2)\right)}{\sum_{k'}\exp\left(\frac{1}{\tau}(f(E_i)/\|f(E_i)\|_2)^T(\mathbf{c}_{k'}/\|\mathbf{c}_{k'}\|_2)\right)}, \quad (9)$$

where $\tau$ is a temperature parameter for $p_i$ and a sharper version for another view $q_i$. The distillation loss can be formulated as:

$$L_{Distill} = -\frac{1}{M}\sum_{i=1}^{M} q_i \log p_i. \quad (10)$$

We also adopt a entropy maximization regularizer [4] for the unsupervised objective, which can be formulated as:

$$L_{ENT} = -\frac{1}{M}\sum_{i=1}^{M} p_i \log p_i, \quad (11)$$

However, we found that maximizing entropy, while encouraging the exploration of new categories, also decreases the model's confidence in known categories, ultimately sacrificing accuracy on known categories (see Sec. 5.6).

To address this issue, we propose an adaptive entropy optimization strategy, aiming for the model to actively explore new categories while maintaining accuracy on known categories. Specifically, we use pseudo-label consistency across four views to decide on entropy optimization. For a sample, on one hand, we generate two pseudo-labels using $p_i$ and its augmented view $q_i$; on the other hand, we generate $p_i^{Aug}$ and $q_i^{Aug}$ as two additional views to obtain two more pseudo-labels using the multimodal augmentation module mentioned earlier, which provides a more challenging perspective as the generated feature, encouraged by adversarial learning, are more diverse and hence more discriminative. We then use a consistency checker to perform consistency checks on the pseudo-labels from these four different views for entropy optimization, which can be formulated as:

$$L_{Adapt} = \begin{cases} \alpha L_{ENT} & \text{if } n = 4 \\ -\beta L_{ENT} & \text{if } n < 4 \end{cases} \quad (12)$$

where $n$ represents the number of consistency for the pseudo-labels, $\alpha$ and $\beta$ are hyperparameters.

Through this strategy, when the model's predictions are completely consistent across different views, we increase the model's confidence in its judgment by minimizing entropy; when there is a discrepancy in the model's judgments across views, we encourage further exploration by maximizing entropy as we want the model to explore new events as much as possible when there is uncertainty, rather than blindly gravitating towards known events.

## 4.5 Overall Formulation and Optimization

In this study, we optimize a minimax problem via a straightforward back-propagation way. To summarize the previous discussions, the overall objective function of DAEO can be formulated as follows:

$$(\hat{\theta}_{CLIP}, \hat{\theta}_f) = \arg\min_{\theta_{CLIP}, \theta_f} L_{CE}^{Aug} + L_{CE} + L_{Distill} + L_{Adapt}, \quad (13)$$

---

**Algorithm 1** DAEO Algorithm

**Input:** Labeled data: $D_L = \{(x_i, y_i)\}_{i=1}^{N}$, unlabeled data: $D_U = \{(x_j)\}_{j=1}^{M}$, the CLIP model, the multimodal augmentation model $G$ and the MLP classifier $f$.

**Output:** Learned model parameters $\hat{\theta}_{CLIP}$, $\hat{\theta}_f$ and $\hat{\theta}_G$.

**while** $t \leq$ **MaxIter**

1: Compute the multi-modal features $E_i$ according to Eq. 1.
2: Compute the augmented multi-modal features $E_i^{Aug}$ according to Eq. 3.
3: Compute the pseudo-labels from four different views.
4: Perform a consistency check for these pseudo-labels and compute the adaptive entropy loss $L_{Adapt}$ according to Eq. 12.
5: Compute the cross-entropy loss $L_{CE}$ and $L_{CE}^{Aug}$ and distill loss $L_{Distill}$ according to Eq. 8, Eq. 4 and Eq. 10, respectively.
6: Optimize the objective in Eq. 13.
7: Recompute the multi-modal features $E_i$ and the augmented multi-modal features $E_i^{Aug}$ according to Eq. 1 and Eq. 3, respectively.
8: Recompute the cross-entropy loss $L_{CE}^{Aug}$, KL divergence loss $L_{KL}$ and consistency loss $L_{Consis}$ according to Eq. 4, Eq. 2 and Eq. 5, respectively.
9: Optimize the objective in Eq. 14.

**end while**

---

$$(\hat{\theta}_G) = \arg\max_{\theta_G} L_{CE}^{Aug} - L_{Consis} - L_{KL}, \quad (14)$$

The detailed algorithm for the DAEO method is presented in Algorithm 1.

## 5 EXPERIMENT

### 5.1 Dataset

*5.1.1 Collection of News Events.* The collection of news events plays a crucial role in generalized news event discovery, demanding a diverse array of relationships among different news events to encompass various possibilities. For instance, when using time as a divider to separate the training and test sets, the relationship between events can be identical, subset, intersecting, or entirely distinct, depending on the type and time of the event. In our paper, we define short-term events, cyclical events, and long-term events, which are designed to cover various event relationships and align with real-world scenarios. Specifically, we collect a variety of events ranging from 2010 to 2023 for each type of event from a crowd-sourced platform Wikipedia, e.g., short-term events include natural, human-made disasters, etc.; cyclical events encompass sports competitions, political elections, etc.; long-term events involve political conflicts, social movements, etc. Ultimately, our collection comprises 66 news events, including 42 short-term events, 13 cyclical events and 11 long-term events.

*5.1.2 Collection and Statistics of the Dataset.* For data collection and statistics, we select Twitter as our primary source due to its extensive user base. We employ event-related hashtags and temporal searches to avoid oversimplification of the task. For long-term events, we sample important sub-events based on Wikipedia, e.g., "Syrian Civil War" containing "Ghouta Chemical Attack", "US

**Table 1: Statistic of the MNED dataset.**

| #Events | #Text | #Image | Average Words | #Language |
|---------|-------|--------|---------------|-----------|
| 66 | 161,350 | 196,543 | 17.645 | 63 |

**Table 2: The division of the MNED dataset in the experiments. '#New' refers to the number of new events.**

| Proportion | Training set | | Test set | | |
|------------|--------------|--------|-----------|--------|------|
| | #Sample | #Event | #Sample | #Event | #New |
| 25% | 32,270 | 27 | 121,013 | 55 | 39 |
| 50% | 64,540 | 43 | 80,675 | 42 | 23 |
| 75% | 96,810 | 56 | 40,338 | 23 | 10 |

Troops Withdrawing from Northern Syria", and so on. Subsequently, we filter out single-modality data samples and manually verified the semantic relevance of the samples for the corresponding event, resulting in a multimodal news event discovery (MNED) dataset of 161,350 samples. Data statistics and sample distribution are shown in Table 1 and Figure 4. Further details are available in Appendix.

*5.1.3 Data Partitioning.* Different from other classification tasks, the generalized news event discovery task inherently involves a temporal dimension. Therefore, we organize all posts chronologically and then split the dataset into training and test sets based on sequential proportions. For this purpose, we divide the dataset into training and test sets by selecting three different time points—corresponding to 25%, 50%, and 75% of the timeline of the collected data—to determine the chronological length of the training set relative to the entire dataset. As for the validation set, we allocate 20% of the training set, chosen randomly across categories. The specifics of this division are summarized in Table 2.

## 5.2 Evaluation Metric

To evaluate our model's performance, we employ a clustering accuracy (ACC) followed by [30]. This metric is calculated as follows:

$$ACC = \max_{p \in P(Y_U)} \frac{1}{M} \sum_{i=1}^{M} 1\{y_i = p(\hat{y}_i)\} \qquad (15)$$

where $P$ represents the set of all possible permutations that align the model's predicted labels $\hat{y}_i$ with the actual ground truth labels $y_i$, utilizing the Hungarian method [11] for optimal matching. We apply this metric across three sets: the complete unlabeled set denoted as "All", a subset called "Known" which contains samples from classes already known to the model, and "New", comprising samples from classes not previously seen by the model.

## 5.3 Implementation Details

We utilize the CLIP ViT-B/16 backbone to train all methods, with fine-tuning the final block and linear projection layer of the text and visual encoders. The SGD optimizer [3] is employed with an initial learning rate of 0.001 and then decayed following a cosine schedule. The models are trained over 100 epochs with a batch size of 128. In alignment with [32], the temperature value for distillation learning is set to 0.1 and the sharper version starts at 0.07, then is

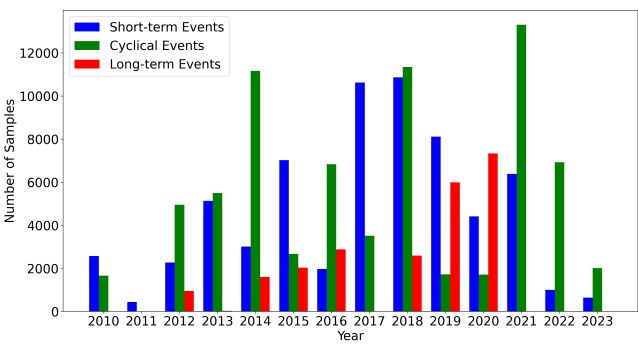

**Figure 4: Distribution of the MNED dataset over time.**

gradually warmed up to 0.04 using a cosine schedule in the starting 10 epochs. The hyperparameters $\alpha$ and $\beta$ are set to 0.03 and 2.3, respectively. For non-English text, we utilize the Google Translation API[1] to translate the content into English. For posts containing multiple images, we only use the first image. The process of tuning and testing is carried out on a separate validation set, facilitating the selection of the best hyperparameters for optimal performance. Additionally, more extended experiments are provided in Appendix.

## 5.4 Baselines

To investigate the effectiveness of our proposed method, we compare it with three different baseline approaches to provide a comprehensive evaluation. 1) K-means method [16]. This baseline extracts the image and text features from the pretrained CLIP model and concatenates them to form the multimodal features, followed by the K-means clustering algorithm. 2) novel category discovery baselines (UNO [7] and RankStats [8]). These are strong baselines from the field of novel category discovery. Following the setup in [30], we configure one classification head to the total number of classes in order to adapt these models to fit the task. 3) the state-of-the-art methods in generalized category discovery (GCD [30] and SimGCD [32]). GCD utilizes semi-supervised K-means clustering based on learned features, and SimGCD employs a parametric classifier for distillation learning, which has demonstrated impressive results across various image recognition tasks.

## 5.5 Comparison with the State of the Arts

Table 3 shows the experimental results of our proposed DAEO method and other comparison methods on the generalized news event discovery task. From these results, we observe that: 1) DAEO outperforms all baselines across most scenarios with different dataset proportions, validating the effectiveness of our model for generalized news event discovery. It's noteworthy that when the training proportion is 25%, our model's performance does not surpass that of the K-means baseline. This is because the relatively small amount of training data, which hinders the model's ability to learn robust event features effectively. In fact, the likelihood of encountering such a limited amount of data is lower in real-world scenarios, especially with the continuous generation of news events. 2) Employing K-means clustering directly on features extracted by the

---

[1]https://cloud.google.com/translate

**Table 3: Results on the MNED dataset.**

| Method | 25% | | | 50% | | | 75% | | |
|---|---|---|---|---|---|---|---|---|---|
| | Known | New | All | Known | New | All | Known | New | All |
| K-means [16] | 0.400 | **0.433** | 0.419 | 0.370 | 0.446 | 0.404 | 0.286 | 0.320 | 0.297 |
| RankStats [8] | 0.275 | 0.050 | 0.147 | 0.441 | 0.165 | 0.320 | 0.461 | 0.446 | 0.457 |
| UNO [7] | 0.736 | 0.176 | 0.419 | 0.792 | 0.277 | 0.567 | 0.735 | 0.450 | 0.647 |
| GCD [30] | 0.464 | 0.333 | 0.390 | 0.463 | 0.435 | 0.451 | 0.450 | 0.441 | 0.447 |
| SimGCD [32] | 0.706 | 0.316 | 0.485 | 0.535 | 0.596 | 0.562 | 0.705 | 0.462 | 0.630 |
| DAEO | **0.757** | 0.409 | **0.560** | **0.823** | **0.622** | **0.735** | **0.735** | **0.629** | **0.702** |
| Δ | +0.052 | +0.093 | +0.075 | +0.287 | +0.024 | +0.172 | +0.029 | +0.167 | +0.072 |

| Backbone | Known | New | All |
|---|---|---|---|
| ViT-B/32 | 0.803 | 0.570 | 0.701 |
| ViT-B/16 | 0.823 | **0.622** | 0.735 |
| ViT-L/14 | **0.841** | 0.611 | **0.740** |

**(a) Backbone.**

| Method | Known | New | All |
|---|---|---|---|
| wo Translation | 0.819 | 0.587 | 0.717 |
| Translation | **0.823** | **0.622** | **0.735** |
| M-CLIP [5] | 0.821 | 0.621 | 0.733 |

**(b) Multilingual processing.**

| Condition | Known | New | All |
|---|---|---|---|
| $n \geq 2$ | **0.826** | 0.465 | 0.668 |
| $n \geq 3$ | 0.825 | 0.552 | 0.706 |
| $n = 4$ | 0.823 | **0.622** | **0.735** |

**(c) Condition for $L_{Adapt}$.**

**Table 4: Ablation experiments with the proportion of 50%.**

**Table 5: Ablation study on the different components of our approach with the proportion of 50%.**

| # | MA | Entmin | Entmax | Adapt | Ctr | Known | New | All |
|---|---|---|---|---|---|---|---|---|
| 1 | × | ✓ | ✓ | ✓ | × | 0.820 | 0.578 | 0.715 |
| 2 | ✓ | × | ✓ | ✓ | × | 0.807 | 0.631 | 0.730 |
| 3 | ✓ | ✓ | × | ✓ | × | 0.742 | 0.220 | 0.513 |
| 4 | ✓ | × | ✓ | × | × | 0.568 | **0.659** | 0.608 |
| 5 | ✓ | × | × | × | × | 0.793 | 0.267 | 0.562 |
| 6 | ✓ | ✓ | ✓ | ✓ | ✓ | 0.804 | 0.620 | 0.723 |
| 7 | ✓ | ✓ | ✓ | ✓ | × | **0.823** | 0.622 | **0.735** |

**Table 6: Results of our approach for different event types with the proportion of 50%.**

| Type | Known | New | All |
|---|---|---|---|
| Short-term Event | 0.891 | 0.642 | 0.656 |
| Cyclical Event | 0.853 | - | 0.853 |
| Long-term Event | 0.662 | 0.527 | 0.600 |

pretrained CLIP model yields impressive results, underscoring the significance of utilizing a robust pretrained model. 3) Parametric learning methods (i.e., SimGCD) outperform non-parametric clustering approaches (i.e., GCD). This is attributed to the joint training of the entire model, which avoids potentially being sub-optimal.

## 5.6 Ablation Study

The proposed DAEO model contains different modules. To validate their effectiveness, we conduct ablation studies on these components. We denote Multimodal Augmentation as "MA", the entropy minimization and maximization terms in $L_{Adapt}$ as "Entmin" and "Entmax", respectively, and "Adapt" to represent $L_{Adapt}$, with "Ctr" indicating self-contrastive learning and supervised contrastive learning. From the results in Table 5, we have the following observations: 1) The absence of multimodal augmentation (#1 and #7) leads to a decrease in accuracy, especially for new events, which underscores the contribution of the multimodal augmentation module in learning more robust features. 2) By comparing #4 and #5, we note that the entropy maximization term boosts the model's performance on recognizing new events but adversely affects its ability to identify known events. The inclusion of $L_{Adapt}$ and Entmin (#4 and #7) not only retains the model's capacity to recognize new events but also improves its accuracy on known events. This demonstrates

the efficacy of the adaptive entropy optimization strategy in balancing the model's performance across known and new events. 3) The addition of self-contrastive learning and supervised contrastive learning (#6 and #7) does not enhance our model's performance, which could be attributed to negative optimization introduced by random augmentations.

In addition, we also investigate the effect of the model's backbone, the handling of multilingual text in the dataset, the conditions for $L_{Adapt}$ and the performance of our model under different event types. 1) As shown in Table 4a, employing larger pretrained models as the backbone improves performance, underscoring the importance of a robust pretrained model. 2) Regarding multilingual text processing, we experiment with using the M-CLIP model [5], which is pretrained on multiple languages. According to the results in Table 4b, utilizing such model does not outperform a straightforward approach of employing the Google Translate API for language translation, thus we select the translation method. 3) For the condition of $L_{Adapt}$, loosening the criteria (i.e., using a consistency threshold across different views to determine entropy minimization/maximization) leads to reduced recognition rates for new events, as shown in Table 4c. Therefore, we select the condition of consistency across all views. 4) As shown in Table 6, the model has a high recognition rate for cyclical events, due to their high degree of similarity and predictable recurrence. However, for long-term events, the ongoing evolution of the events makes the recognition of even known events as challenging as that of short-term events.

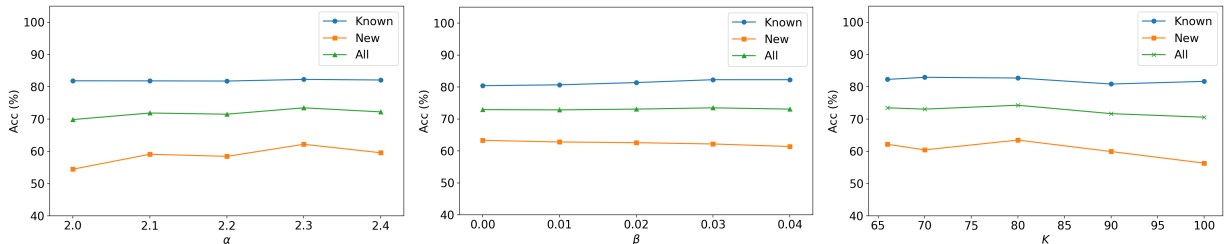

Figure 5: Parameter sensitivity on the MNED dataset with the proportion of 50%.

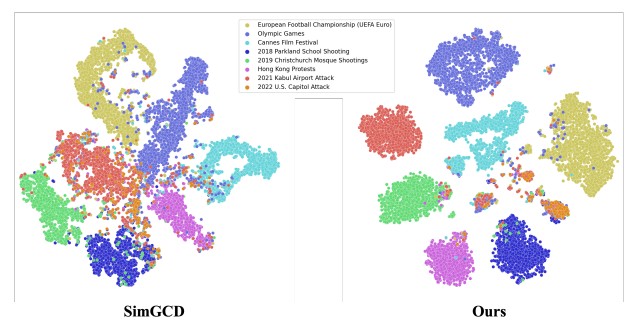

Figure 6: TSNE visualization of multimodal features from selected news events with the proportion of 50%.

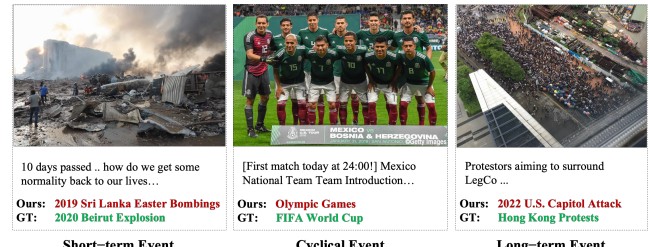

Figure 7: Failure examples of DAEO on the MNED dataset with the proportion of 50%.

## 5.7 Parameter Analysis

To delve into the impact of parameters $\alpha$ and $\beta$ on the model's performance, we conducted experiments varying $\alpha$ from 2.0 to 2.4 and $\beta$ from 0 to 0.04. As depicted in Figure 5, we observe that an increase in $\alpha$ tends to enhance the accuracy for new events at the expense of slightly reducing accuracy for known events, while $\beta$ exhibits an inverse relationship. Overall, the model demonstrates moderate sensitivity to these parameters, leading to the selection of $\alpha = 2.3$ and $\beta = 0.01$ as the optimal settings.

Regarding the parameter $K$, which denotes the number of prototypes, we assume that the number of events is known following [32] in our model. We investigate its effect on our model using different values. As shown in Figure 5, although the performance on new events slightly decreases with increasing $K$ values, the fluctuation remains minimal, which shows our model's robustness to variations in the number of prototypes.

## 5.8 Data Visualization

To further investigate the effectiveness of our proposed method, we employ t-SNE [29] visualization to illustrate the multimodal event features learned by the model. We select eight similar events, including both known and new events, for visualization. As shown in Figure 6, we have the following observations: 1) Compared to SimGCD, the features from our proposed method have clearer boundaries between different events, which proves the effectiveness of our approach. 2) For similar events, such as attack events, our model demonstrates a strong capability to differentiate between them, which is attributed to our multimodal augmentation module that utilizes adversarial learning to generate discriminative features.

## 5.9 Case Study

Despite the excellent performance of DAEO, Figure 7 shows three failure cases from different event types. We observe that their misclassification mainly stems from the lack of distinctive elements in the provided images and texts. Specifically, the first case comes from the "2020 Beirut Explosion". Due to the absence of key information about the Beirut explosion, the event is mistakenly classified as a general explosion event. The second case, "FIFA World Cup", included a team photo, which is common in other sports events, such as the "Olympic Games". The third case, "Hong Kong Protests", featured many protesting people, leading the model to mistakenly categorize it as an attack event. These failure cases illustrate the complexity and challenges of the generalized news event discovery.

## 6 CONCLUSION

In this paper, we introduce a Dynamic Augmentation and Entropy Optimization (DAEO) model designed specifically to tackle the challenges of generalized news event discovery. A multimodal augmentation module is designed to employ adversarial learning to generate distinctive multimodal features, which improves the model's ability to discern between similar event categories. An adaptive entropy optimization strategy with a self-distillation method leverages pseudo-labels from different views to adaptively optimize entropy, thereby enhancing the model's ability in recognizing both new and known events. Additionally, we contribute to the field by introducing the Multimodal News Event Discovery (MNED) dataset, which contains various event types and serves as a valuable resource for researchers. Extensive experiments conducted on the MNED dataset validate the effectiveness of our proposed model.

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
