# OpenReview forum: "Generalized News Event Discovery via Dynamic Augmentation and Entropy Optimization"
_acmmm.org/ACMMM/2024/Conference — MM2024 Poster_

### Official Review · Reviewer_qv8m · 2024-05-25

**Rating:** 4
**Confidence:** 3

**Summary:**

The paper introduces the Dynamic Augmentation and Entropy Optimization (DAEO) model for news event discovery on social media, capable of identifying both known and new events using multimodal data. It features an adversarial learning-based augmentation module and an adaptive entropy optimization strategy with self-distillation. The researchers also compiled the Multimodal News Event Discovery (MNED) dataset, containing 161,350 samples of 66 real-world events. Experiments on this dataset demonstrate the model's effectiveness, offering a significant contribution to multimedia applications and data integration innovations.

**Strengths:**

1.	Comprehensive Work: The paper provides a thorough and detailed study. The model design incorporates multiple modules to effectively process news event data. Additionally, the authors have developed a dedicated dataset and performed relevant partitioning (varying proportions and different types), demonstrating substantial engineering effort.
2.	Effective Model Results: The paper mentions that the experimental results on the MNED dataset demonstrate the effectiveness of the proposed model. This indicates that the model performs well on the dataset provided.
3.	Strong Logical Structure: The paper is logically structured with a clear problem background, module introduction, and experimental analysis. Overall, the writing is coherent, and the content is sufficiently detailed.

**Limitations:**

1.	Baseline Selection Needs Improvement: From the results presented in Table 3 (particularly for 25% and 50%), it is evident that some baselines perform particularly poorly. The authors' analysis of the experimental results is also insufficient. For instance, the authors did not analyze the results for RankStats and UNO, despite describing them as strong baselines in the field of novel category discovery. However, their performance in the "New" category is significantly worse than others.
2.	Motivation for Dataset Creation Needs Clarification: The paper mentions that temporal information is important in the dataset but does not elaborate on the motivation behind this. The authors divided the dataset into Short-term, Cyclic, and Long-term events and split it based on time. However, the paper merely presents the corresponding results without explaining the rationale behind this approach. The motivation and benefits of this division should be clearly articulated.
3.	Design of Augmentation Modules: In section 4.3, the authors mention that applying random data augmentation on single modalities does not enhance model performance and choose to use feature-level augmentation. However, in section 4.4, they augment directly from augmented images and original texts, which differs from the method in the previous section. The choice of these two different augmentation methods seems inconsistent even though they are in different parts of the model.

**Suitability:**

2

---

### Official Review · Reviewer_viHJ · 2024-05-25

**Rating:** 4
**Confidence:** 3

**Summary:**

This paper introduces a groundbreaking approach to News Event Discovery (NED) that addresses the challenge of detecting both familiar and novel events in a multimodal context.  Departing from conventional assumptions, the authors propose a Dynamic Augmentation and Entropy Optimization (DAEO) framework, designed to cope with the dynamic nature of real-life news occurrences.

**Strengths:**

The idea of the division of different events makes sense for generalized news event discovery. And the design of the MNED dataset can better enrich the research in this field.

**Limitations:**

1.	Descriptive Contradiction: The details in figure 2 and figure 3 are contradictory, and the description in METHODOLOGY is a little vague for figures, so it is difficult for readers to understand all the detail of the method proposed by author according to the description and figures.
2.	Ablation Study: In Table 4a, the author may consider utilizing larger pretrained models as the backbone, such as Vit-H, due to its broader application and superior performance.
3.	Reproduction: Will the authors open-source the code they implement? It is important for the reproduction of this paper and further research.

**Suitability:**

2

---

### Official Review · Reviewer_zi5J · 2024-05-27

**Rating:** 3
**Confidence:** 3

**Summary:**

The paper studies a challenging issue of unknown events to news discovery and proposes a Dynamic Augmentation and Entropy Optimization (DAEO) model for the task. The proposed DAEO introduces a multimodal augmentation module and an adaptive entropy optimization strategy to improve multimodal representation and uncover new events.

**Strengths:**

1. The paper studies a significant and challenging issue in news event discovery and proposes a self-motivated DAEO model with dynamic augmentation and adaptive entropy optimization.
2. The paper is well-organized and easy to follow.
3. The paper delivers a new dataset MNED for news event discovery, contributing to the event discovery community.

**Limitations:**

1. The notation delta in Table 3 is explained and confusing.
2. The paper fails to detail the previous work on discovering new events and analyses the correlations between the paper and the previous studies.
3. The paper lacks experiments to interpret or visualize the capability of the proposed model in handling new news events. Such experiments are beneficial for understanding the benefits and contributions of the proposed model.

**Suitability:**

3

---

### Official Review · Reviewer_XscT · 2024-06-06

**Rating:** 6
**Confidence:** 2

**Summary:**

This paper introduces the Dynamic Augmentation and Entropy Optimization (DAEO) model for generalized news event discovery, which aims to identify both known and new events from multimodal social media data. The key contributions include:

1. Formulation of the generalized news event discovery problem and the collection of a comprehensive Multimodal News Event Discovery (MNED) dataset.

2. A multimodal augmentation module that generates robust augmented features using adversarial learning to distinguish between similar events.

3. An adaptive entropy optimization strategy that selectively optimizes entropy based on multi-view pseudo-label consistency to balance known and new event recognition.

4. A self-distillation learning framework that trains a unified prototypical classification head for all classes using pseudo-labels from different views.

**Strengths:**

This paper introduces a novel framework for essentially topic recognition and discovery, which is a fairly important task in multimedia discovery. The authors integrated two important modules, namely the multimodal augmentation and the adaptive entropy optimization module. Their introduction in fairly well justified, especially for the AEO module, which seems to be the highlight of this work. Although EO is conventionally, the authors have introduced an additional adaptivity which balances between known and unknowed features. All modules are also well integrated via the adversarial learning paradigm. Finally, the collection and hopefully eventual publication of the collected dataset should also be useful for future researchers.

**Limitations:**

This is a relatively solid work and I don't have much complaints about it in general; the only lacking aspect which can be mentioned is the lack of comparison with an alternative dataset, which makes the presented results slightly thin; hopefully the authors can address this shortcoming in the rebuttal?

**Suitability:**

3

---

### Meta-Review · Area_Chair_k2uC · 2024-07-02

**Recommendation:** Accept (Poster)
**Confidence:** 5

**Metareview:**

News event Discovery is a highly subjective area of research.  The area chair is not surprised that the paper got such complementary feedback from the reviewers. Overall, the area chair recommends the paper as a poster.